# Nearly half of adults with symptoms of sexually transmitted infections (STIs) did not seek clinical care: A population-based study of treatment-seeking behavior among adults in Rakai, Uganda

Yasmin P. Ogale [1]*, Caitlin E. Kennedy[1], Fred Nalugoda[2], Josephine Mpagazi[2], Jade C. Jackson[3], Ronald Galiwango[2], Robert Ssekubugu[2], Godfrey Kigozi[2], Julie A. Denison[1], Charlotte A. Gaydos[3], Joseph Kagaayi[2], M. Kathryn Grabowski[3]

1 Johns Hopkins Bloomberg School of Public Health, Baltimore, MD, United States of America, 2 Rakai Health Sciences Program, Rakai, Uganda, 3 Johns Hopkins School of Medicine, Baltimore, MD, United States of America

⊙ These authors contributed equally to this work.

* Yasmin.ogale@gmail.com

**Data Availability Statement:** A deidentified version of the Rakai Community Cohort Study data can be

## Abstract

Understanding treatment-seeking behavior is critical to the treatment and control of sexually transmitted infections (STIs), yet current data on STI treatment seeking in low-resource settings is rare. This population-based study aimed to describe STI treatment-seeking behavior and identify factors associated with seeking treatment at a clinic among adults with STI-related symptoms in rural Uganda. The STI prevalence study (STIPS) conducted a survey and STI testing among all consenting adults aged 18–49 in two communities in rural south-central Uganda. Of 1,825 participants, 962 individuals self-reported STI symptoms in the past six months; we present descriptive data on treatment seeking and STI prevalence among these individuals. We used multivariable Poisson regressions with robust variance to determine the sociodemographic and symptom-related factors independently associated with seeking STI treatment at a clinic and assessed the association with previous clinic treatment seeking and current STI diagnosis. Forty-three percent of adults who reported STI-related symptoms in the past six months said they did not seek any treatment. Among those who did, 58% sought treatment at a private clinic, 28% at a government clinic, 9% at a pharmacy/drug store, 3% at a traditional healer, 2% at a market/shop, and 5% at another location. Among both males and females, having multiple STI related symptoms was positively associated with clinic treatment seeking (males = PRR: 1.73, 95%CI: 1.36–2.21; females = PR: 1.41, 95%CI: 1.12–1.78). Approximately one-third of males and females who reported previously seeking clinic treatment for their symptoms were diagnosed with a curable STI at the time of the survey. In this setting, nearly half of adults with STI-related symptoms are not seeking clinical care and many who report having sought treatment for recent STI symptoms have curable STIs. Future studies should explore barriers to care-seeking and strategies to improve STI services.

provided to interested parties subject to completion of the Rakai Health Sciences Program data request form and signing of a Data Transfer Agreement. Inquiries should be directed to datarequests@rhsp. org.

**Funding:** This work was supported by grants from the Johns Hopkins Center for AIDS Research (MKG; grant number P30AI094189), grants from the National Institute of Allergy and Infectious Diseases (MKG; grant numbers R01AI143333, R01MH115799, and K01AI125086), the National Institute of Mental Health (MKG; grant number R01MH107275), the Eunice Kennedy Shriver National Institute of Child Health and Human Development (MKG; grant number RO1HD091003), and the Division of Intramural Research of the National Institute for Allergy and Infectious Diseases (MKG). The funders had no role in study design, data collection and analysis, decision to publish, or preparation of the manuscript.

**Competing interests:** The authors have declared that no competing interests exist.

# Introduction

Understanding STI treatment-seeking behavior is critical to STI control given that timely and appropriate STI treatment can reduce duration of infectiousness and limit the spread of infection [1, 2]. Treatment-seeking behavior can be defined as any activity undertaken by an individual who perceives themselves to be ill or have a health problem, in order to find a remedy. Various factors can influence the decision to seek treatment, including individual-level factors such as symptom recognition, preexisting beliefs and meanings of symptoms, and perceived efficacy of different treatment methods [1, 2]. Contextual factors such as the availability, quality, and accessibility of treatment, as well as social stigma, also play a role [1, 2]. These factors influence not only the timing and location of treatment seeking, but also whether an individual seeks treatment at all [1, 2].

Despite its importance, population-based data on STI treatment seeking in low-resource settings is scarce. In such settings, syndromic management by a healthcare worker is used to diagnose presumptive infection [3, 4], and so seeking clinical care is the best way to receive effective treatment. Data from the 2016 Uganda DHS show that, of those who reported having an STI or STI symptoms, 71% of women and 64% of men sought advice or treatment from a clinic, hospital, private doctor or other health professional, while 26% of women and 34% of men did not seek any advice or treatment [5]. Previous studies on STI treatment-seeking behavior in Uganda report a variety of treatment approaches, including public health facilities, private practitioners, traditional healers and self-treatment [6, 7]. A qualitative study from 1999 on STI treatment-seeking behaviors among Ugandan adults found that treatment in the informal sector, including self-treatment and traditional healers, was especially common (over 60% of participants). The study also found that for participants with STI-related symptoms deterrents to seeking care at public health facilities included long waiting times, lack of drugs, user fees, corruption and bribes by health workers, health workers abusing STI patients, lack of privacy, long distances, fear of being tested for AIDS, specimens not being examined in the laboratory and being given tablets instead of injections (which were preferred) [6].

Previous studies in the region have also observed lower levels of STI treatment seeking in women, with barriers to care including lack of access to facilities, lack of time and money, and dependence on men for permission to leave the home, which resulted in women either ignoring their problem, using self-care or self-medication, using herbal or traditional medicine, or using services in secret [8–13]. Consultations with husbands, relatives and friends also delayed treatment seeking for women, with prompt care triggered by symptoms that were perceived to be more severe or life threatening [14]. Notably, gender disparities in STI care stand in contrast to what is typically observed for HIV care and treatment where men are less likely to engage in HIV services than women [15].

While useful in providing context, previous studies from low-resource settings, including those mentioned above, are limited in that many sampled participants at treatment facilities only after they presented for care; most studies do not capture individuals who delay seeking treatment or who do not seek treatment at all. Here, we used a population-based sample to describe treatment-seeking behavior among adults with STI-related symptoms in rural Uganda and identify factors associated with seeking treatment at a clinic.

# Methods

## Ethics statement

STIPS was approved by the Uganda Virus Research Institute Research Ethics Committee (GC/ 127/19/03/709) and the Johns Hopkins School of Medicine Institutional Review Board

(IRB00204691). The study was also registered with the Ugandan National Council for Science and Technology (HS 364 ES).

## Study setting

This study was nested in the STI prevalence study (STIPS), a population-based survey conducted in the Rakai region of south-central Uganda that aimed to estimate population STI burden [16]. STIPS was conducted as a part of the Rakai Community Cohort Study (RCCS), one of the oldest population-based studies of HIV in Africa. Conducted by the Rakai Health Sciences Program (RHSP), the RCCS is an ongoing, open community-based cohort of residents aged 15–49 years in agrarian communities, semi-urban trading centers and Lake Victoria fishing communities in the Rakai region. The RCCS includes the administration of a demographic and health questionnaire, as well as HIV testing for all consenting participants, every 18 to 24 months. Details of the RCCS study design can be found elsewhere [17].

## Data collection

STIPS was a population-based survey that recruited all individuals aged 18–49 years from two RCCS communities—one inland and one fishing—from May to October 2019 [16]. These communities were selected because of their representativeness, geographical diversity, population size, and enrollment timelines within the RCCS survey. Three adjacent villages consisting of semi-urban and rural agrarian populations comprised the inland community and one fish landing site along Lake Victoria comprised the fishing community. There were 2583 census-eligible participants, of whom 1884 were present at the time of RCCS survey, 1877 consented to participate in RCCS survey and 1825 consented to participate in STIPS.

In addition to the standard RCCS questionnaire, STIPS participants were administered an STI module that assessed their symptom status and treatment-seeking behavior. To ascertain symptom status, each participant was prompted on a list of symptoms and asked to identify each symptom that they had experienced in the past six months (previous symptoms) and also in the past 7 days (recent symptoms). Symptoms included: genital ulcer, genital discharge, frequent urination, painful urination, pain during intercourse, bleeding during intercourse, lower abdominal pain and genital warts, as well as thick and/or colored vaginal discharge, itching of the vagina and unpleasant vaginal odor for females. Treatment-seeking behavior was defined as self-reported treatment seeking from any location for STI-related symptoms in the past six months, and was assessed among participants who reported experiencing previous or recent symptoms: participants were asked if they did anything to help cure their symptoms or to prevent passing on infection to their spouse or partner(s), and if so, what action(s) did they take: used condoms, abstinence, sought treatment for self, sought treatment for partner or some other action. Those who reported seeking treatment for themselves were also asked to specify where they went for treatment: pharmacy/drug store, market/shop, Rakai Program Clinic (RHSP-run clinic providing HIV care and related services, comparable to private clinics), government doctor/nurse/clinic, private doctor/nurse/clinic, traditional healer or other. Up to three locations were recorded per participant.

In addition to the routine HIV screening conducted in the RCCS, STIPS participants were also evaluated for Chlamydia trachomatis (CT), Neisseria gonorrhoeae (NG), Trichomonas vaginalis (TV) and syphilis. All consenting participants provided genital swabs at the time of interview for testing (clinician-collected penile urethral meatus swabs for males and self-administered vaginal swabs for females). CT/NG testing was performed using the Abbott Real-Time CT/NG assay using the Abbott m2000 RealTime System for PCR testing at the RHSP central laboratory. TV testing was performed using the OSOM Trichomonas Rapid Test (Sekisui) at the time of the survey. Syphilis screening was performed using the SD Bioline 3.0, a

solid phase immunochromatographic assay for the qualitative detection of antibodies of all iso-types (IgG, IgM, IgA) against T. pallidum. Syphilis screening was performed with HIV testing at time of survey; the rapid plasma reagin test (RPR) was then performed within 24 hours at the RHSP central laboratory for all participants with positive screening results to determine syphilis titers. All assays were conducted according to the manufacturers' protocol. All individuals who tested positive for any STI were provided treatment by RHSP per the Ugandan National Clinical Treatment Guidelines for STIs.

## Data analysis

We first measured the overall prevalence of STI-related symptoms among all STIPS participants. For the remaining analyses, we restricted our sample to only those who reported any STI-related symptoms in the past six months, including those reporting symptoms in the past 7 days. First, we conducted a descriptive analysis of the data, including assessing the prevalence of previous STI symptoms in the past six months. Second, we measured the prevalence of treatment seeking overall and at each treatment location (e.g. government clinic, private clinic, pharmacy/drug store) and compared the prevalence of private versus government clinic treatment seeking. Third, we compared sociodemographic and symptom-related factors among participants who sought treatment at a government or private clinic (clinic) compared to those who sought treatment at a non-clinic location (e.g. pharmacy/drug store, market/shop, traditional healer, etc.) or who sought no treatment at all. Finally, we compared the prevalence of any curable STI (NG, CT, TV or active syphilis) between participants who did and did not previously seek clinic treatment, and assessed the univariable association between previous clinic treatment seeking and current prevalence of curable STIs. All prevalence risk ratios were estimated with modified Poisson regression models with robust variance [18]. Given the different social and economic contexts of males and females, we conducted analyses for the sample as a whole, as well as for each sex separately. We also stratified data for each sex by community type.

Complete treatment seeking information (i.e. any treatment seeking [yes or no] and specific treatment location) was collected for 99.4% of the sample, with only 6 participants dropped from the analysis because of missing treatment data. Age was analyzed in five-year age groupings. Consistent condom use was assessed within non-marital relationships only; marital relationships were excluded because condom use is rare within these partnerships. We also calculated the number of symptoms in the past week, and in the past six months, as a sum of a participant's self-reported symptoms in the respective time period. We assessed positive STI diagnoses with the STIPS test result, with syphilis RPR titers $\geq$8 considered indicative of high titer syphilis infection [19].

Informed by Kroeger's determinants model [1, 20] and McLeroy et al.'s socioecological framework [21], we created a conceptual framework for treatment-seeking behavior (S1 Fig). Using this framework and prior literature, we selected variables for inclusion in the final multivariable models. Given that this study was conducted in two communities, we only adjusted for relevant variables at the individual level including number of STI-related symptoms experienced in the past six months, as well as select sociodemographic characteristics (age, community type, marital status, HIV status). All data analysis was carried out in STATA version 15 [22].

## Results

### Sociodemographic characteristics, sexual behavior and STI symptomatology

Fifty-three percent (962/1,825) of STIPS participants reported any STI symptoms in the six months prior to their interview date (34% [290/860] of males; 70% [672/964] of females).

Sixty-three percent (605/962) of those with symptoms in the past six months reported at least one symptom in the past seven days (51% [149/290] of males; 68% [456/672] of females). Table 1 summarizes the demographic characteristics of STIPS participants reporting STI symptoms. Most males were aged 30–39 years, in a monogamous marriage and from the fishing community. Most females were aged 20–29 years and also in a monogamous marriage. Approximately one-third of those with STI symptoms were people living with HIV (PLHIV). With respect to sexual behaviors, approximately half of the males reported 2–4 sexual partners in the past year (148 [51%]) and the majority reported 5–10 sexual partners in their lifetime (205 [71%]). Over three-quarters of females reported having one sexual partner in the past year (529 [79%]) and just over half of females reported 2–4 lifetime sexual partners (408 [61%]). Consistent condom use with non-marital partners was uncommon.

In terms of symptoms, the majority of males reported only one symptom (161 [56%]) in the past six months, with painful urination being the most common (42%). The majority of females reported 2–4 symptoms (342 [51%]) in the past six months, with vaginal itching being the most common (62%) (Fig 1). Half (149 [51%]) of males reported experiencing symptoms in the seven days before the interview as compared to 68% (456) of females. Of those who reported symptoms in the past seven days, most males reported one symptom in the past week (101 of 149 [68%]) and most females reported 2–4 symptoms in the past week (208/456 [46%]).

## Treatment seeking for STI symptoms

Overall, 545 participants (57%) reported seeking any treatment for their symptoms, while the remaining 43% reported seeking no treatment at all (Table 1). The likelihood of seeking any treatment was similar between males and females (160/290 [55%] males; 385/671 [57%] females; PRR: 1.04, 95% CI: 0.92–1.18). The prevalence of seeking any treatment also did not differ by community type (PRR: 1.04, 95% CI: 0.93–1.16). No significant differences were observed when further stratifying by sex and community type.

Fig 2 shows where participants sought treatment for their symptoms. Nearly all participants (95%, 515/541) who sought treatment did so at only one location, while 5% (26/541) reported seeking treatment at two locations. Of those who sought treatment, 58% sought treatment at a private clinic, 28% at a government clinic, 9% at a pharmacy/drug store, 3% at a traditional healer, 2% at a market/shop, and 5% at some other location. Private clinics were the most commonly reported treatment location (71% among males; 53% among females). S1 Table describes treatment seeking at private or government clinics by sex and community type. Females were more likely to seek treatment at government clinics than males in both communities (S1 Table). Individuals in the fishing community were significantly more likely to seek treatment at a private clinic than those in the inland community irrespective of sex (S1 Table).

## Treatment seeking at a clinic

A total of 457 participants (48%) reported seeking treatment at a clinic (government or private) for their symptoms. There was no difference in the prevalence of clinic treatment by sex (47% males; 48% females; PRR: 1.02, 95% CI: 0.88–1.18) or by community type (48% inland; 47% fishing; PRR: 0.98, 95% CI: 0.78–0.86). When stratifying community type, however, we found that males in the fishing community were significantly less likely to seek clinic treatment than males in the inland community (55% males in inland; 43% males in fishing; PRR: 0.78, 95% CI: 0.61–1.00).

S2 Table shows associations between demographic and behavioral factors and STI treatment seeking (S3 Table presents these data stratified by sex). Overall, of males who sought

**Table 1. Sociodemographic characteristics and symptomatology of STIPS participants who reported STI symptoms in the past 6 months (N = 962), by sex.** Data are presented as n (%).

| | Total | Male | Female |
|---|---|---|---|
| | N = 962 | N = 290 | N = 672 |
| Age | | | |
| 15–19 years | 69 (7%) | 14 (5%) | 55 (8%) |
| 20–29 years | 371 (39%) | 98 (34%) | 273 (41%) |
| 30–39 years | 356 (37%) | 115 (40%) | 241 (36%) |
| 40–49 years | 166 (17%) | 63 (22%) | 103 (15%) |
| Marital status | | | |
| Never Married | 75 (8%) | 33 (11%) | 42 (6%) |
| Married, Monogamous | 560 (58%) | 174 (60%) | 386 (57%) |
| Married, Polygamous | 128 (13%) | 23 (8%) | 105 (16%) |
| Previously Married | 199 (21%) | 60 (21%) | 139 (21%) |
| Community type | | | |
| Inland | 420 (44%) | 96 (33%) | 324 (48%) |
| Fishing | 542 (56%) | 194 (67%) | 348 (52%) |
| Religion (N = 935) | | | |
| Christian | 806 (86%) | 244 (84%) | 562 (87%) |
| Muslim | 123 (13%) | 45 (16%) | 78 (12%) |
| Other/none | 6 (1%) | 1 (0%) | 5 (1%) |
| Education | | | |
| None | 62 (6%) | 21 (7%) | 41 (6%) |
| Some primary education or above | 900 (94%) | 269 (93%) | 631 (94%) |
| Occupation | | | |
| Agricultural or housework | 342 (36%) | 50 (17%) | 292 (43%) |
| Bar or restaurant work | 71 (7%) | 1 (0%) | 70 (10%) |
| Boda boda driving or trucking | 20 (2%) | 20 (7%) | 0 (0%) |
| Fishing | 128 (13%) | 128 (44%) | 0 (0%) |
| Student | 12 (1%) | 4 (1%) | 8 (1%) |
| Trader or shopkeeper | 220 (23%) | 36 (12%) | 184 (27%) |
| Other | 169 (18%) | 51 (18%) | 118 (18%) |
| HIV status (N = 960) | | | |
| Negative | 649 (68%) | 200 (69%) | 449 (67%) |
| Positive | 311 (32%) | 90 (31%) | 221 (33%) |
| Sex in the past year | | | |
| No | 55 (6%) | 9 (3%) | 46 (7%) |
| Yes | 907 (94%) | 281 (97%) | 626 (93%) |
| Sexual partners in the past year | | | |
| None | 55 (6%) | 9 (3%) | 46 (7%) |
| 1 | 631 (66%) | 102 (35%) | 529 (79%) |
| 2–4 | 243 (25%) | 148 (51%) | 95 (14%) |
| 5–10 | 25 (3%) | 24 (8%) | 1 (0%) |
| >10 | 8 (1%) | 7 (2%) | 1 (0%) |
| Sex with partner from outside the community | | | |
| No | 711 (74%) | 162 (56%) | 549 (82%) |
| Yes | 251 (26%) | 128 (44%) | 123 (18%) |
| Lifetime sexual partners | | | |
| None | 10 (1%) | 2 (1%) | 8 (1%) |

*(Continued)*

**Table 1.** (Continued)

| | Total | Male | Female |
|---|---|---|---|
| | **N = 962** | **N = 290** | **N = 672** |
| 1 | 77 (8%) | 1 (0%) | 76 (11%) |
| 2–4 | 460 (48%) | 52 (18%) | 408 (61%) |
| 5–10 | 376 (39%) | 205 (71%) | 171 (25%) |
| >10 | 39 (4%) | 30 (10%) | 9 (1%) |
| Condom use in past 12 months | | | |
| Marital partners only | 544 (57%) | 102 (35%) | 442 (66%) |
| Non-marital partners, inconsistent or no use | 325 (34%) | 152 (52%) | 173 (26%) |
| Non-marital partners, consistent use | 61 (6%) | 31 (11%) | 30 (4%) |
| NA, no sexual partners in past 12 months | 32 (3%) | 5 (2%) | 27 (4%) |
| Number of STI symptoms in past 6 months | | | |
| 1 | 313 (33%) | 161 (56%) | 152 (23%) |
| 2–4 | 462 (48%) | 120 (41%) | 342 (51%) |
| > = 5 | 187 (19%) | 9 (3%) | 178 (26%) |
| Symptoms in the past 7 days | | | |
| No | 357 (37%) | 141 (49%) | 216 (32%) |
| Yes | 605 (63%) | 149 (51%) | 456 (68%) |
| Number of STI symptoms in past 7 days | | | |
| 0 | 357 (37%) | 141 (49%) | 216 (32%) |
| 1 | 261 (27%) | 101 (35%) | 160 (24%) |
| 2–4 | 250 (26%) | 42 (14%) | 208 (31%) |
| > = 5 | 94 (10%) | 6 (2%) | 88 (13%) |
| Sought treatment for STI symptoms (N = 961) | | | |
| No | 416 (43%) | 130 (45%) | 286 (43%) |
| Yes[a] | 545 (57%) | 160 (55%) | 385 (57%) |

[a]Includes government clinics, private clinics, RHSP clinics, pharmacies/drug stores, markets/shops, traditional healers, herbalists, or other location.

clinic treatment, most were aged 20–39 years, married in a monogamous union, and working in the fishing industry. Of females who sought clinic treatment, most were aged 20–29 years, married in a monogamous union, and engaged in agriculture or housework. Thirty-three percent of men and 35% of females who sought treatment at a clinic were living with HIV. Females were less likely to seek treatment if they had no sexual partners in the past year (PRR: 0.59, 95% CI: 0.37–0.94).

Table 2 describes STI symptomatology for males and females who reported seeking clinic treatment. Painful urination (52%), genital discharge (47%) and genital ulcers (35%) were the most common symptoms reported among males who sought treatment at a clinic. Vaginal itching (69%), genital discharge (57%) and vaginal discharge (50%) were the most frequently reported symptoms among females who sought clinic treatment. About half of males who reported seeking treatment at a clinic reported 2–4 STI symptoms in the past six months (73/136 [54%]). This was similar for females (157/321 [49%]).

Among those who sought clinical care for their symptoms in the last six months, 46% of males (63/136) and 66% of females (211/321) reported having at least one STI symptom in the past week. Furthermore, among males and females who reported at least one symptom in the past week, just over half reported that they did not seek treatment at a clinic in the past six months for their symptoms (85/148 [57%] and 244/455 [54%], respectively).

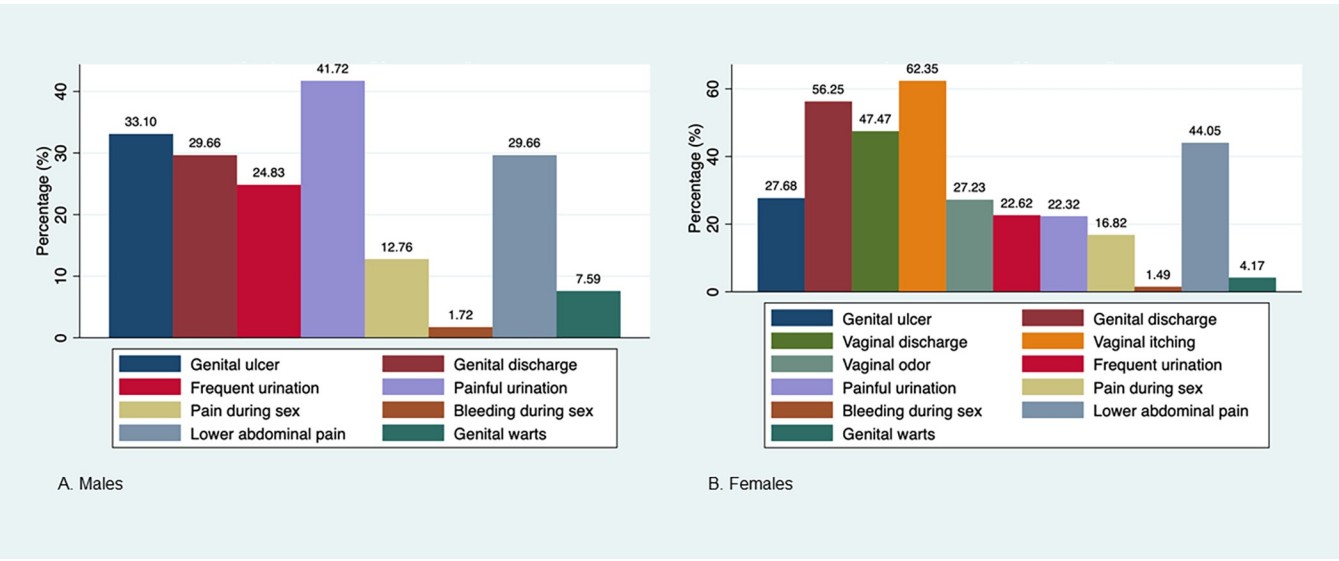

**Fig 1. Symptomatology\* of STIPS participants who reported STI-related symptoms in past 6 months (N = 962), by sex.** Frequency of self-reported STI-related symptoms experienced in the past six months among (A) males and (B) females. \*Multiple response options were possible.

The number of STI symptoms reported in the past six months and the number of STI symptoms reported in the past week were also associated with clinic treatment seeking for both males and females. Certain symptoms were more strongly associated with clinic treatment seeking (Table 2). For example, the likelihood of seeking treatment in a clinic for males increased with reporting of painful urination in the past six months (PRR: 1.55, 95% CI: 1.22–1.98) and more than doubled with reporting of genital discharge (PRR: 2.16, 95% CI: 1.73–2.69).

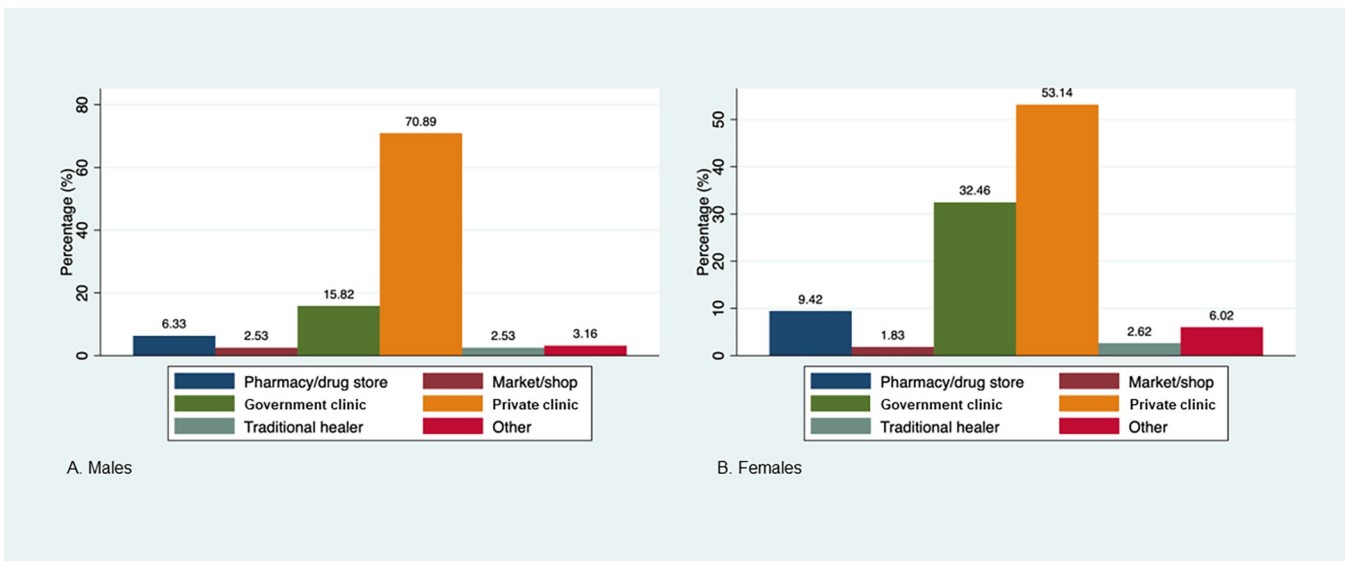

**Fig 2. Treatment locations\* reported by STIPS participants who sought treatment for their symptoms in the past 6 months, by sex (N = 545).** Frequency of various treatment locations sought by (A) males and (B) females who self-reported seeking treatment for their STI-related symptoms in the past six months. \*Multiple response options were possible.

**Table 2. Unadjusted prevalence of clinic treatment-seeking among STIPS participants who reported STI symptoms in the past 6 months (N = 956), STI symptomatology for the full sample and by sex.** Data are presented as n (%).

| | FULL SAMPLE (N = 956) | | MALES (N = 288) | | FEMALES (N = 668) | |
|---|---|---|---|---|---|---|
| | Sought Clinic Treatment | | Sought Clinic Treatment | | Sought Clinic Treatment | |
| | Yes N = 457 | Crude PRR (95% CI) | Yes N = 136 | Crude PRR (95% CI) | Yes N = 321 | Crude PRR (95% CI) |
| Number of STI symptoms in past 6 months | | | | | | |
| 1 | 112/312 (36%) | REF | 58/161 (36%) | REF | 54/151 (36%) | REF |
| 2–4 | 230/457 (50%) | 1.40\*\*\* (1.18–1.67) | 73/118 (62%) | 1.72\*\*\* (1.34–2.21) | 157/339 (46%) | 1.30\*\* (1.02–1.65) |
| > = 5 | 115/187 (61%) | 1.71\*\*\* (1.42–2.06) | 5/9 (56%) | 1.54 (0.83–2.87) | 110/178 (62%) | 1.73\*\*\* (1.36–2.20) |
| Any STI symptoms in the past 7 days | | | | | | |
| No | 183/353 (52%) | REF | 73/140 (52%) | REF | 110/213 (52%) | REF |
| Yes | 274/603 (45%) | 0.88\*\* (0.77–1.00) | 63/148 (43%) | 0.82 (0.64–1.04) | 211/455 (46%) | 0.90 (0.76–1.06) |
| Number of STI symptoms in past 7 days | | | | | | |
| 0 | 183/353 (52%) | REF | 73/140 (52%) | REF | 110/213 (52%) | REF |
| 1 | 100/260 (38%) | 0.74\*\*\* (0.62–0.89) | 35/100 (35%) | 0.67\*\*\* (0.49–0.92) | 65/160 (41%) | 0.79\*\* (0.63–0.99) |
| 2–4 | 121/249 (49%) | 0.94 (0.80–1.10) | 24/42 (57%) | 1.10 (0.81–1.49) | 97/207 (47%) | 0.91 (0.75–1.10) |
| > = 5 | 53/94 (56%) | 1.09 (0.89–1.33) | 4/6 (67%) | 1.28 (0.71–2.30) | 49/88 (56%) | 1.08 (0.86–1.35) |
| Genital ulcer | | | | | | |
| No | 295/675 (44%) | REF | 88/192 (46%) | REF | 207/483 (43%) | REF |
| Yes | 162/281 (58%) | 1.32\*\*\* (1.16–1.51) | 48/96 (50%) | 1.09 (0.85–1.40) | 114/185 (62%) | 1.44\*\*\* (1.23–1.68) |
| Genital discharge | | | | | | |
| No | 210/495 (42%) | REF | 72/204 (35%) | REF | 138/291 (47%) | REF |
| Yes | 247/461 (54%) | 1.26\*\*\* (1.11–1.44) | 64/84 (76%) | 2.16 (1.73–2.69) | 183/377 (49%) | 1.02 (0.87–1.20) |
| Thick and/or colored vaginal discharge | | | | | | |
| No | 296/638 (46%) | REF | - | - | 160/350 (46%) | REF |
| Yes | 161/318 (51%) | 1.09 (0.95–1.25) | - | - | 161/318 (51%) | 1.11 (0.95–1.30) |
| Itching of the vagina | | | | | | |
| No | 236/539 (44%) | REF | - | - | 100/251 (40%) | REF |
| Yes | 221/417 (53%) | 1.21\*\*\* (1.06–1.38) | - | - | 221/417 (53%) | 1.33\*\*\* (1.11–1.59) |
| Unpleasant vaginal odor | | | | | | |
| No | 363/774 (47%) | REF | - | - | 227/486 (47%) | REF |
| Yes | 94/182 (52%) | 1.10 (0.94–1.29) | - | - | 94/182 (52%) | 1.11 (0.93–1.31) |
| Frequent urination | | | | | | |
| No | 343/733 (47%) | REF | 110/217 (51%) | REF | 233/516 (45%) | REF |
| Yes | 114/223 (51%) | 1.09 (0.94–1.27) | 26/71 (37%) | 0.72\* (0.52–1.01) | 88/152 (58%) | 1.28\*\*\* (1.09–1.51) |
| Painful urination | | | | | | |
| No | 287/687 (42%) | REF | 65/169 (38%) | REF | 222/518 (43%) | REF |

*(Continued)*

**Table 2.** (Continued)

| | FULL SAMPLE (N = 956) | | MALES (N = 288) | | FEMALES (N = 668) | |
|---|---|---|---|---|---|---|
| | Sought Clinic Treatment | | Sought Clinic Treatment | | Sought Clinic Treatment | |
| | Yes N = 457 | Crude PRR (95% CI) | Yes N = 136 | Crude PRR (95% CI) | Yes N = 321 | Crude PRR (95% CI) |
| Yes | 170/269 (63%) | 1.51*** (1.33–1.72) | 71/119 (60%) | 1.55*** (1.22–1.98) | 99/150 (66%) | 1.54*** (1.32–1.79) |
| Pain during intercourse | | | | | | |
| No | 370/806 (46%) | REF | 115/251 (46%) | REF | 255/555 (46%) | REF |
| Yes | 87/150 (58%) | 1.26*** (1.08–1.48) | 21/37 (57%) | 1.24 (0.91–1.69) | 66/113 (58%) | 1.27*** (1.06–1.52) |
| Bleeding during intercourse | | | | | | |
| No | 453/941 (48%) | REF | 134/283 (47%) | REF | 319/658 (48%) | REF |
| Yes | 4/15 (27%) | 0.55 (0.24–1.29) | 2/5 (40%) | 0.84 (0.29–2.49) | 2/10 (20%) | 0.41 (0.12–1.43) |
| Lower abdominal pain | | | | | | |
| No | 259/577 (45%) | REF | 95/203 (47%) | REF | 164/374 (44%) | REF |
| Yes | 198/379 (52%) | 1.16** (1.02–1.33) | 41/85 (48%) | 1.03 (0.79–1.34) | 157/294 (53%) | 1.22*** (1.04–1.42) |
| Genital warts | | | | | | |
| No | 443/906 (49%) | REF | 131/266 (49%) | REF | 312/640 (49%) | REF |
| Yes | 14/50 (28%) | 0.57** (0.37–0.90) | 5/22 (23%) | 0.46*** (0.21–1.01) | 9/28 (32%) | 0.66 (0.38–1.14) |

*** p≤0.01

** p≤0.05

* p≤0.1; NA: Not applicable

In multivariable analyses, significant factors associated with seeking treatment at a clinic for males included being from the inland community and having multiple STI-related symptoms in the past six months. For females, the only significant factor associated with seeking STI treatment at a clinic was having multiple STI-related symptoms in the past six months (S4 Table).

### Previous treatment seeking and current STI prevalence

Among those who reported STI-related symptoms, CT prevalence was 11%, NG was 10%, TV was 13%, and active syphilis was 7%. Among individuals who reported previously seeking clinic treatment for their symptoms, approximately one-third tested positive for any curable STI (CT, NG, TV or active syphilis) at time of survey (45/136 [33%] males; 98/321 [31%] females). We found no significant difference in the current prevalence of curable STIs between those who did and did not previously seek clinic treatment, for either sex (Table 3).

## Discussion

This population-based study evaluated STI-related symptoms and treatment seeking in rural Uganda and examined factors associated with seeking treatment at a government or private clinic. Just over half (57%) of adults who reported STI-related symptoms in the past six months reported seeking any treatment for their symptoms, with similar rates in males and females. Seeking treatment at a clinic was 48% overall (47% among males; 48% among females). While our estimates were based on a population-based sampling approach of all eligible adults in our study communities, our estimations of clinic treatment seeking are lower than those found in

**Table 3. Prevalence of any curable STI (CT, NG, TV or active syphilis) at the time of the survey among STIPS participants who reported STI symptoms in the past 6 months (N = 956), by sex.**

| Sex | Sought Clinic Treatment | Any curable STI<br>Yes N = 299 | Crude PRR (95% CI) |
|---|---|---|---|
| Males (N = 288) | No | 41/152 (27%) | REF |
| | Yes | 45/136 (33%) | 1.23 (0.86–1.75) |
| Females (N = 668) | No | 115/347 (33%) | REF |
| | Yes | 98/321 (31%) | 0.92 (0.74–1.15) |

\*\*\* p≤0.01, \*\* p≤0.05, \* p≤0.1

previous studies, including the 2016 Ugandan DHS (70% for the country) and the first round of the RCCS in the 1990s (67% for the region) [5, 23].

Our data showed no difference in the prevalence of clinic treatment seeking by sex. This was unexpected, as research in Uganda and other resource-limited settings suggest differential patterns of treatment seeking [11, 24–30], with females more likely to delay, and less likely to seek, STI treatment than males [31–34]. While the 2016 Uganda DHS estimates show a higher prevalence of clinic treatment seeking among females than males [5], as aforementioned, we are hesitant to compare our findings to DHS estimates because of urban/rural differences. We also found that sexual behaviors which may influence STI risk, such as having multiple partners and types/locations of sexual partners (i.e. household, community, extra-community partners), differed by sex, as has been explored further in other publications from the RCCS [35, 36]. We recommend that future researchers continue to assess sex-specific treatment-seeking behavior across a range of settings.

We also found that males in the fishing community were significantly less likely to seek clinic treatment, as compared to males in the inland community. The epidemiologic differences between fishing and inland communities in Rakai are well-established: fishing communities show a disproportionate burden of HIV, high prevalence of sexual risk behaviors and historically showed a lower use of combination HIV prevention services (though this has been increasing in recent years with significant new service provision) [36–39]. Furthermore, data show that overall, males in Rakai are less likely to be enrolled in HIV care [40, 41], as are in-migrants [40]. Rakai fishing communities, or 'landing sites', generally have a high proportion of males, the majority of whom migrate from other communities in order to work as fishermen on Lake Victoria. Assuming that the barriers to HIV care for Rakai residents–including stigma, demanding work schedules, transport costs, belief in spiritual healing, long wait times and inadequate staff respect for patients [42]–also apply to STI treatment seeking, our observations indicating a lower prevalence of clinic treatment seeking among males in the fishing community is not unexpected. What is surprising, however, is the fact that this difference was not observed among females. We recommend future researchers explore the intersection of sex and community type, as well as try to better understand the barriers to treatment seeking in fishing communities among males.

In addition, we found the presence of some, but not all, symptoms to be associated with clinic treatment seeking for each sex. For instance, lower abdominal pain, pain or bleeding during intercourse, and genital ulcers showed no association with treatment seeking among males in our study. The same goes for genital warts and thick/colored discharge among females. Not recognizing STI-related symptoms, not perceiving them as severe, or not attributing them to STI-related causes can prevent treatment seeking [1, 14, 43, 44]. A similar phenomenon may have occurred in our sample: participants may not have attributed lower

abdominal pain, warts or discharge to an STI, thereby explaining why the presence of some symptoms were associated with clinic treatment over others. For females, symptoms such as lower abdominal pain, dysuria, and frequent urination can also be due to urinary tract infections, which are common among females, and which may help explain why females reported symptoms more often than males in our study.

The frequency of private clinic use in our sample was notable. The Ugandan healthcare system suffered losses during the decades of civil unrest in the 1980s [45]. Consequentially, many Ugandans have come to perceive health centers as expensive and lacking medication, and often turn to self-medication first and use health centers as a last resort [45–49]. Also as a result of the political turmoil, the number of public health services in the Uganda decreased and the number of private clinics increased [49, 50]. While we did not include a treatment location mapping exercise in our study, we did find a 2010 study that mapped the availability of private and public facilities in rural areas of Uganda [50]. Based on their work, the authors reported that public facilities made up 4.3% of all the health care units that were mapped as compared to private facilities which made up 95.7%. Private-for-profit clinics and drug shops made up 17.1% of all mapped facilities and private-not-for-profit facilities made up 1.6% of all mapped facilities [51]. While still considered rural, the districts included in their study are more central and developed than Rakai district. Nevertheless, we expect that their finding of more private than public clinics may still apply to Rakai. As such, the high prevalence of private clinic treatment that we observed may be partially explained by the high availability of private clinics in the area.

We also found a difference in private versus government clinic treatment seeking by sex, with females more likely to attend government clinics than males as seen in other settings [52–54]. A lack of finances, unfriendly reception and long wait times have been identified as reasons for why females do not seek care at formal sector clinics [54].

The lack of association between clinic treatment seeking and HIV status surprised us; we expected that individuals who were HIV-seropositive to be more likely to seek clinic treatment for their symptoms than those who were HIV-negative. This was observed among Rwandan women [55]. Eighty-six percent of PLHIV in our study reported currently taking antiretroviral treatment (ART); we would have expected that their routine interaction with the health system due to ART would make them more likely to attend a clinic for treatment than those without HIV. PLHIV may also be more conscious of their sexual risk behavior and sexual health than those who are negative [56–58], furthering their likelihood of seeking clinic treatment. We speculate that the availability and affordability of services or stigma may have affected treatment-seeking behavior among PLHIV and recommend further research in this area.

We also found that approximately one-third of males and females who previously sought clinic treatment for their symptoms were diagnosed with at least one curable STI (CT, NG, TV or active syphilis) at the time of the survey. Furthermore, our analysis showed no difference in the current prevalence of curable STIs comparing those who previously sought clinic treatment versus those who did not, for either sex. Assuming that seeking clinic treatment meant receiving treatment, these data could indicate that reinfection rates were high, treatment was inadequate, or both. A discussion on STI prevalence and asymptomatic infection in STIPS communities regardless of treatment seeking is explored elsewhere [16]; further studies exploring the temporal association between past treatment seeking, including receiving and adhering to treatment, and current STI prevalence are recommended to assess treatment effectiveness.

A strength of our study lies in its population-based sample, which is rare in other studies focused on treatment-seeking behavior. Calls have been made for a broader research perspective in order to understand sexual healthcare seeking behavior [59]. This perspective includes a focus on non-attendance at healthcare services as well as research that uses non-patient

samples recruited from non-medical settings in order to accurately capture the range of behaviors, perspectives, and health issues occurring within the population and ensure appropriate and effective service provision [35]. We addressed these items in our study by interviewing all eligible individuals in our study communities and including both persons who did and did not seek treatment. Taken together, this information can provide program managers and decision-makers with a better understanding of treatment-seeking patterns within the community.

However, our research is not without limitations. First, our estimations of clinic treatment seeking may be lower than national estimates because of the communities included in our sample. The communities that we included are considered rural and treatment seeking tends to be lower in rural settings [1, 31, 60]. Second, we did not further define treatment settings or ask participants to name specific treatment locations. As a result, some treatment locations may have been misclassified (e.g. drug shops being reported as private clinics, etc.). Third, inclusion of study participants was based on self-reported symptom history in the past six months. It is possible that eligible participants may not have been included because they were too shy, or embarrassed, to share their symptom history (social desirability bias), did not remember their symptoms (recall bias) or did not understand the terms we used for the symptoms. Fourth, given that the survey was administered by RHSP staff, it is possible that participants over-reported seeking clinical care or under-reported seeking care in informal sectors (or not seeking any care at all) in order to please the interviewer. Given the lower than expected rates of treatment seeking, however, we doubt this was a major concern. Fifth, our adjusted models did not consider all factors associated with treatment seeking, such as environmental, social, psychosocial, economic, geographic or service-related factors, as well as symptom severity, which have been shown to be associated with treatment-seeking behavior in low-resource settings [1]. We also did not measure the availability of each type of treatment location in our study communities–this information would be useful to contextualize our results. Sixth, we note that, while our study was sufficiently powered to assess differences between males and females, we may have been underpowered to detect differences by both sex and community type. This sub-analysis may be of programmatic interest; we recommend that researchers consider this when designing future studies.

Finally, our analysis grouped together private and government clinics, and compared them to other treatment locations/no treatment. In doing so, we assumed that these clinics provided appropriate and effective care, of equal/sufficient quality, and that anything but clinical care was ineffective. This may be an unfair assumption: in low-resource settings, formal sector facilities often show poor quality of care in general [61, 62]. A study specifically on the quality of STI case management in Ugandan private clinics and drug shops concluded that the quality of management was poor [45]. We recommend studying quality of care and barriers to providing quality care in local facilities, as well as urge leadership to strengthen care and enforce quality standards across health service sectors and facilities.

## Conclusion

Timely and appropriate diagnosis is critical to STI treatment and control. We found that half of adults with STI symptoms in two rural Ugandan communities are not seeking appropriate clinical care under the syndromic management strategy. We recommend researchers continue to focus on treatment-seeking behavior in low-resource settings and explore barriers to seeking care, including health system barriers such as low quality of care. However, as many people with STIs are asymptomatic in this and other settings, we also urge decision-makers to increase support for STI services, as well as increased availability of diagnostics, in this and similar contexts.

## Supporting information

**S1 Fig. Conceptual framework.** Informed by Kroeger's determinants model and McLeroy et al.'s socioecological framework. This conceptual framework was developed to guide statistical analyses. Variables selected for inclusion in the multivariate model are underlined.
(TIF)

**S1 Table. Prevalence of private and government clinic treatment seeking among STIPS participants who reported seeking clinic treatment for STI symptoms in the past 6 months (N = 450), by sex and community type.** Data are presented as n (%).
(DOCX)

**S2 Table. Unadjusted prevalence of clinic treatment seeking among STIPS participants who reported STI symptoms in the past 6 months (N = 956), sociodemographic characteristics and behaviors for the full sample.** Data are presented as n (%).
(DOCX)

**S3 Table. Unadjusted prevalence of clinic treatment seeking among STIPS participants who reported STI symptoms in the past 6 months (N = 956), sociodemographic characteristics and behaviors by sex.** Data are presented as n (%).
(DOCX)

**S4 Table. Predictors of clinic treatment seeking among STIPS participants who reported STI symptoms in the past 6 months (N = 956), by sex.**
(DOCX)

## Acknowledgments

We are grateful to the community members and leaders of Rakai who participated in this research. We also give special appreciation to all the RHSP staff members and in particular the STIPS team for supporting this research. We would also like to acknowledge the support of Drs. Ronald Gray and Maria Wawer for their support for this project undertaken at Rakai, and Joyce Yehjin Jang and Ping Teresa Yeh for their help preparing this manuscript for submission.

## Author Contributions

**Conceptualization:** Yasmin P. Ogale, Joseph Kagaayi, M. Kathryn Grabowski.

**Data curation:** Yasmin P. Ogale.

**Formal analysis:** Yasmin P. Ogale.

**Funding acquisition:** M. Kathryn Grabowski.

**Methodology:** Joseph Kagaayi, M. Kathryn Grabowski.

**Project administration:** Josephine Mpagazi, Jade C. Jackson, Ronald Galiwango, Godfrey Kigozi, Joseph Kagaayi, M. Kathryn Grabowski.

**Resources:** Caitlin E. Kennedy, Godfrey Kigozi, Joseph Kagaayi, M. Kathryn Grabowski.

**Software:** Robert Ssekubugu.

**Supervision:** Caitlin E. Kennedy, Joseph Kagaayi, M. Kathryn Grabowski.

**Validation:** Yasmin P. Ogale, Robert Ssekubugu.

**Writing – original draft:** Yasmin P. Ogale.

**Writing – review & editing:** Yasmin P. Ogale, Caitlin E. Kennedy, Fred Nalugoda, Josephine Mpagazi, Jade C. Jackson, Ronald Galiwango, Robert Ssekubugu, Godfrey Kigozi, Julie A. Denison, Charlotte A. Gaydos, Joseph Kagaayi, M. Kathryn Grabowski.

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
