## [Decision Letter · Decision Letter 0]

6 Mar 2023

PGPH-D-23-00150

Nearly half of adults with symptoms of sexually transmitted infections (STIs) did not seek clinical care: A population-based study of treatment-seeking behavior among adults in Rakai, Uganda

Dear Dr. Ogale ,

Thank you for submitting your manuscript to PLOS Global Public Health. After careful consideration, we feel that it has merit but does not fully meet PLOS Global Public Health’s publication criteria as it currently stands. Therefore, we invite you to submit a revised version of the manuscript that addresses the points raised during the review process.

We look forward to receiving your revised manuscript.

Kind regards,

Peter Bai James, PhD

Academic Editor

Journal Requirements:

3. Please provide separate figure files in .tif or .eps format only and remove any figures embedded in your manuscript file. Please also ensure that all files are under our size limit of 10MB.

4. We have noticed that you have uploaded Supporting Information files, but you have not included a list of legends. Please add a full list of legends for your Supporting Information files after the references list.

Additional Editor Comments (if provided):

Reviewers' comments:

Reviewer's Responses to Questions

**Comments to the Author**

1. Does this manuscript meet PLOS Global Public Health’s publication criteria? Is the manuscript technically sound, and do the data support the conclusions? The manuscript must describe methodologically and ethically rigorous research with conclusions that are appropriately drawn based on the data presented.

Reviewer #1: Yes

Reviewer #2: Yes

2. Has the statistical analysis been performed appropriately and rigorously?

Reviewer #1: Yes

Reviewer #2: Yes

3. Have the authors made all data underlying the findings in their manuscript fully available (please refer to the Data Availability Statement at the start of the manuscript PDF file)?

Reviewer #1: No

Reviewer #2: Yes

4. Is the manuscript presented in an intelligible fashion and written in standard English?

Reviewer #1: Yes

Reviewer #2: Yes

5. Review Comments to the Author

Reviewer #1: Overall

The manuscript by Ogale et al. presents important information on treatment seeking behaviour for STIs in Uganda. As discussed by the authors, this is a relatively under-researched area as most studies enrol participants when they attend for care. The introduction was informative and concise, and the methodology appears to be sound and was generally very clear, although more information should be provided on the conceptual framework underlying the multivariate analysis. Although the results section was adequately presented, I did find the large number of analyses quite overwhelming. Although I understand the rationale behind presenting results for male and female participants separately, the reader has to process double the number of results, and this should be taken into consideration when deciding on which results to present. At present, the results section is very long. Despite this, the discussion was thorough and did an excellent job of summarising the results and placing it in the context of the literature, particularly regarding HIV and fishing communities, as well as some of the limitations of the study.

Overall, I think the results section could be made more concise. Otherwise, I have suggested mostly minor edits for clarity.

Introduction

1. Lines 95-102: “Previous studies in the region have also found gender differences in treatment-seeking, with barriers to care including lack of access to facilities, lack of time and money, and dependence on men for permission to leave the home, resulting in women either ignoring their problem, using self-care or self medication, using herbal or traditional medicine, or using services in secret [8-13]. Consultations with husbands, relatives and friends would also delay treatment-seeking for women, with prompt care triggered by symptoms that were perceived to be more severe or life threatening [14].”

I found this paragraph really interesting, as it contrasted with my assumptions and experiences regarding engagement with healthcare. The referenced articles support reduced levels of care-seeking in women in the region, however there is also a significant evidence base that supports that male engagement with healthcare is particularly problematic (this is often in the context of the HIV care cascade, e.g. described in the UNAIDS report “Male engagement in HIV testing, treatment and prevention in Eastern and Southern Africa” and other literature)

This is not essential, but the authors could consider briefly mentioning that, at least in other contexts, poor male engagement with healthcare is often more prominent. Could this perhaps have contributed to the lack of a gender difference in treatment-seeking?

Methods

2. Lines 121-122: “The RCCS includes the administration of a demographic and health questionnaire, as well as HIV testing for all consenting participants.”

A very minor point, but consider including how often each follow-up round is in the RCCS.

3. Lines 175-176: “For the remaining analyses, we restricted our sample to only those who reported any STI-related symptoms in the past six months (n=962).”

For clarity, please consider stating whether this includes or excludes participants who only reported “recent symptoms” in the past 7 days, as either option is arguably valid.

4. Lines 204-208: “Based on a conceptual framework of treatment-seeking behavior [1], we included in our model the number of STI-related symptoms experienced in the past six months, as well as select sociodemographic characteristics (age, community type, marital status, HIV status) that could theoretically affect treatment-seeking behavior.”

The reference for the conceptual framework is from a 1997 paper by Ward et al. The main model described in the paper by Ward et al. is the “determinants model”, itself described by Kroeger in 1983. Given how old these references are, I am wondering if any more up to date references could be provided that guided your choice of model variables. Additionally, a visual depiction of your framework/pathway would be really helpful, perhaps as a supplementary appendix?

Results

5. Lines 217-219: “Most men were aged 30-39 years, in a monogamous marriage, from the fishing community, Christian, educated at some level and working in the fishing industry.” & Table 1

A definition of “educated at some level” would be helpful. For example, does one year of schooling count as “educated at some level” or completion primary school or another cut-off?

6. Table 1: Condom use

Was condom use asked of married people, or is it assumed that condoms were not used in marriage? If the former, I suggest providing this information. If the latter, please state this in the methods.

7. Lines 265-268: “Comparing private and government clinics, women were less likely to seek treatment at private clinics (more likely to seek government clinics) than men (PRR: 0.76, 95% CI: 0.68-0.85). This trend was seen in both the fishing (PRR: 0.86, 95% CI: 0.76-0.96) and inland (PRR: 0.65, 95% CI: 0.51-0.83) communities. When comparing men across communities, men in fishing communities were significantly more likely to seek treatment at a private clinic than men in inland communities (PRR: 1.25, 95% CI: 1.04-1.51). Similarly, women in the fishing community were significantly more likely to seek treatment at private clinics than women in the inland community (PRR: 1.65, 95% CI: 1.36-2.01).”

There is a huge amount of information in this paragraph, and one has to really concentrate to process it. I think the two main points are:

1. Women are more likely to seek treatment at government clinics than men (both communities)

2. Individuals in fishing communities are more likely to seek treatment at private clinics than in inland communities (both men and women)

Please consider condensing or re-wording this paragraph to be more concise.

8. Lines 302-306: “Just over half of men who reported seeking care at a clinic for their symptoms reported no symptoms in the past week (73/136 [54%]). In contrast, the majority of women who reported seeking care at a clinic for their symptoms reported having at least one STI symptom in the past week (66%), with 65/321 [20%] reporting one symptom 97/321, [30%] reporting 2-4 symptoms and 49/321 [15%] reporting five or more symptoms.”

Minor point: Consider re-phrasing to present the proportion of men who reported seeking care who reported at least one symptom in the past week, as this is how the equivalent statistic is presented for women. i.e. “Just under half of men who reported seeking care at a clinic for their symptoms reported at least one symptom in the past week…” This should make comparability between men and women slightly easier, without having to consider the inverse for men.

9. Table 2

This table provides lots of information. However, as the “clinic” and “no clinic” columns are the inverse of each other, you could consider removing one of these columns so that the table is more concise, and a bit less overwhelming.

10. Table 2

Minor point – For “Number of STI symptoms in past 6 months” for males, the odds ratio for 2-4 is 1.72 (1.34 – 2.21), but there is no associated ** or ***

11. Lines 340 – 342: “In multivariable analyses, significant factors associated with seeking treatment at a clinic for men included being from the inland community and having multiple STI-related symptoms in the past six months.” & Table 3

Please see previous comment regarding the conceptual framework. More information needs to be provided in the methods on the conceptual framework and how that led you to decide to include or exclude variables.

Table 4

Similar comment re: table 2 about redundancy of some data presented. As “yes” and “no” for “any curable STI” are the inverse of each other, I suggested removing the “no” column.

Reviewer #2: The study by Ogale et al. reports on treatment seeking behaviours for sexually transmitted infections in rural communities from Uganda. The authors report that while STI symptoms are common, a considerable proportion of people do not seek treatment from formal healthcare (or at all). In addition, among those with STI symptoms the prevalence of STIs was very high. This study is very relevant to understanding the context for STIs in this setting (and likely more generally in communities across sub-Saharan Africa) and can contribute to developing strategies for improving SRH services.

Comments

Methods: study setting:

- please add the catchment population size to provide more context to the data

- describe how the 2 communities included in this study were selected from the entire population.

Results:

- a flow diagram reporting on inclusion would be helpful in providing an overview of people eligible to participate, those included, those interviewed, and those tested for STIs. This can be added in the supplement

Discussion

- The discussion could be improved by commenting on people with STIs who are asymptomatic and who contribute to transmission (given that a considerable proportion of people who have positive tests for STIs are in fact asymptomatic) and on the importance of the availability of diagnostics for STIs given the high prevalence in these communities and the limitations of syndromic management. The latter is only very briefly mentioned in the conclusion and would deserve a somewhat more extensive comment. If possible it would be interesting to also comment on the prevalence of STIs (positive tests) in Uganda or in other countries from sub-Saharan Africa to provide more context to the high prevalence reported here.

- Another point that can be discussed is that for women (who reported symptoms more frequently than men) symptoms considered as suggestive of STIs included lower abdominal pain, dysuria, and frequency. These symptoms can also be due to urinary tract infections which are common in women.

- It would also be interesting to comment on gender differences in terms of number of sexual partners (women having fewer) and types of sexual partners (more likely to be from inside the community).

Tables and figures: Figure 2: consider writing “private” in full rather than abbreviated.

Abstract line 61 – missing „for“

6. PLOS authors have the option to publish the peer review history of their article (what does this mean?). If published, this will include your full peer review and any attached files.

**Do you want your identity to be public for this peer review?** For information about this choice, including consent withdrawal, please see our Privacy Policy.

Reviewer #1: No

Reviewer #2: **Yes**

---

## [Decision Letter · Decision Letter 1]

4 Apr 2023

Nearly half of adults with symptoms of sexually transmitted infections (STIs) did not seek clinical care: A population-based study of treatment-seeking behavior among adults in Rakai, Uganda

PGPH-D-23-00150R1

Dear, Dr Ogale

We are pleased to inform you that your manuscript 'Nearly half of adults with symptoms of sexually transmitted infections (STIs) did not seek clinical care: A population-based study of treatment-seeking behavior among adults in Rakai, Uganda' has been provisionally accepted for publication in PLOS Global Public Health.

Best regards,

Peter Bai James, PhD

Academic Editor

Reviewer Comments (if any, and for reference):

Reviewer's Responses to Questions

**Comments to the Author**

1. If the authors have adequately addressed your comments raised in a previous round of review and you feel that this manuscript is now acceptable for publication, you may indicate that here to bypass the “Comments to the Author” section, enter your conflict of interest statement in the “Confidential to Editor” section, and submit your "Accept" recommendation.

Reviewer #1: All comments have been addressed

Reviewer #2: All comments have been addressed

2. Does this manuscript meet PLOS Global Public Health’s publication criteria? Is the manuscript technically sound, and do the data support the conclusions? The manuscript must describe methodologically and ethically rigorous research with conclusions that are appropriately drawn based on the data presented.

Reviewer #1: Yes

Reviewer #2: Yes

3. Has the statistical analysis been performed appropriately and rigorously?

Reviewer #1: Yes

Reviewer #2: Yes

4. Have the authors made all data underlying the findings in their manuscript fully available (please refer to the Data Availability Statement at the start of the manuscript PDF file)?

Reviewer #1: No

Reviewer #2: Yes

5. Is the manuscript presented in an intelligible fashion and written in standard English?

Reviewer #1: Yes

Reviewer #2: Yes

6. Review Comments to the Author

Reviewer #1: All comments have been addressed.

Reviewer #2: Thank you for the revision of the manuscript. I have no further comments.

7. PLOS authors have the option to publish the peer review history of their article (what does this mean?). If published, this will include your full peer review and any attached files.

**Do you want your identity to be public for this peer review?** For information about this choice, including consent withdrawal, please see our Privacy Policy.

Reviewer #1: **Yes: **Dr Kevin Martin

Reviewer #2: **Yes: **Ioana D Olaru
